# Homologs of Ancestral CNNM Proteins Affect Magnesium Homeostasis and Circadian Rhythmicity in a Model Eukaryotic Cell

**DOI:** 10.3390/ijms24032273

**Published:** 2023-01-23

**Authors:** Sergio Gil, Helen K. Feord, Gerben van Ooijen

**Affiliations:** 1School of Biological Sciences, University of Edinburgh, Max Born Crescent, Edinburgh EH9 3BF, UK; 2Helmholtz German Research Centre for Geosciences, Telegrafenberg, 14473 Potsdam, Germany

**Keywords:** circadian clock, magnesium transport, CNNM, transporter proteins, *Ostreococcus tauri*

## Abstract

Biological rhythms are ubiquitous across organisms and coordinate key cellular processes. Oscillations of Mg^2+^ levels in cells are now well-established, and due to the critical roles of Mg^2+^ in cell metabolism, they are potentially fundamental for the circadian control of cellular activity. The identity of the transport proteins responsible for sustaining Mg^2+^ levels in eukaryotic cells remains hotly debated, and several are restricted to specific groups of higher eukaryotes. Here, using the eukaryotic minimal model cells of *Ostreococcus tauri*, we report two homologs of common descents of the Cyclin M (CNNM)/CorC protein family. Overexpression of these proteins leads to a reduction in the overall magnesium content of cells and a lengthening of the period of circadian gene expression rhythms. However, we observed a paradoxical increase in the magnesium content of the organelle fraction. The chemical inhibition of Mg^2+^ transport has a synergistic effect on circadian period lengthening upon the overexpression of one CNNM homolog, but not the other. Finally, both homologs rescue the deleterious effect of low extracellular magnesium on cell proliferation rates. Overall, we identified two CNNM proteins that directly affect Mg^2+^ homeostasis and cellular rhythms.

## 1. Introduction

Most organisms have developed an internal mechanism known as the circadian clock that allows for the physiological synchronization to, and anticipation of, 24 h environmental changes in light and temperature that result from the Earth’s rotation [1]. Clock-driven rhythmicity involves the rhythmic expression of a large percentage of genes, driven by clock genes that engage in transcriptional translational feedback loops (TTFLs). 

In addition, the past few years have seen the (re)emergence of studies of non-transcriptional circadian timekeeping in the absence of TTFL activity in a divergent set of organisms [2]. One study reported the discovery that intracellular levels of Mg^2+^ describe circadian oscillations in eukaryotic cells [3]. Mg^2+^ is the most abundant divalent cation within the cell and is crucial for cellular biochemistry, including Mg^2+^-dependent enzyme function in every metabolic pathway and in every step that requires ATP as a source of energy [3,4,5,6]. There are even direct implications of Mg^2+^ rhythms during sleep/wake cycles in humans or photosynthesis in plants pointing to Mg^2+^ as a potential meta-regulator of metabolic state [5,7,8]. However, the molecular mediators of these daily rhythms, as well as the cellular processes through which these ion rhythms are generated, have not been elucidated. 

Mg^2+^ homeostasis is sustained by the activity and/or abundance of specific transmembrane proteins embedded in cellular membranes [3]. Mg^2+^ transport proteins have been documented in many different organisms, but remain largely understudied compared with proteins transporting Ca^2+^, K^+^, or Na^+^ [9,10,11]. While some of these proteins are lineage-specific, such as mammalian TRPM7 [12], many eukaryotic Mg^2+^-transporting proteins have an ancestral prokaryotic homolog [13]. It is therefore likely that the mediators of Mg^2+^ transport that are shared between eukaryotic species are of prokaryotic origin.

The conserved protein families of common descent are SLC41/MgtE, CNNM/CorC and MRS2/CorA [14,15]. Recent studies have directly implicated SLC41/MgtE in Mg^2+^ transport and the maintenance of daily Mg^2+^ rhythms and cellular regulation in several eukaryotic organisms [3,5,15]. CNNM proteins are crucial for Mg homeostasis in many eukaryotes [16], but their exact cellular function in Mg^2+^ homeostasis is still controversial. CNNM proteins are conserved in nearly all organisms, from bacteria to mammals [17,18]. Despite the structural diversity among eukaryotic and prokaryotic CNNMs, they all have a transmembrane Domain of Unknown Function 21 (DUF21) and a pair of cystathionine β-synthase (CBS) domains responsible for regulating CNNM activity by binding the Mg–ATP complex through Mg^2+^ binding sites [18]. Some homologs, including prokaryotic proteins, have a transport-related CorC-HlyC domain near the C-terminus. On the other hand, some eukaryotic homologs contain a phosphatase of the regenerating liver (PRL) protein, which forms heterodimers with the CBS domains of CNNM proteins to regulate Mg^2+^ efflux [19]. 

In this study, we explored the role of CNNM proteins in the circadian clock of a well-established model cell for circadian rhythms, *Ostreococcus tauri*. This microalga displays robust circadian rhythms with a reduced genome and has extensively been used to study conserved regulatory mechanisms of cellular rhythmicity. We identified two CNNM homologs that directly affect circadian rhythmicity, as well as overall cellular Mg^2+^ homeostasis. While our experiments clearly demonstrate divergence of function between these two proteins, both CNNMs improve growth under Mg^2+^ depletion. Our data clearly identify two Mg^2+^ transporters from the CNNM protein family that directly modulate biological rhythms and fine-tune cellular Mg^2+^ levels.

## 2. Results

### 2.1. Homologs of CNNM Proteins in Ostreococcus tauri

Firstly, to identify homologs in the *Ostreococcus tauri* proteome, the Pfam DUF21 consensus sequence was used to BLAST against its proteome. Using this method, two proteins were identified in the *Ostreococcus tauri* proteome: *Ot*CNNM1 (ostta05g01490) and *Ot*CNNM2 (ostta11g01030) (Figure 1A). In addition to the DUF21 and CBS-pair domains, CNNM1 presents a cyclic nucleotide–monophosphate binding domain (cNMP bd) commonly found in ion channels and kinases, both in eukaryotes and prokaryotes [20]. CNNM2 has a C-terminal CorC-HlyC ion transport domain. 

We included reference proteomes of representative species for other eukaryotic clades. After removing all redundant isoforms, we built a phylogenetic tree with all identified sequences. This analysis revealed two clearly distinct phylogenetic clades (Figure 1B). Clade A includes *Ot*CNNM1 and all human proteins (*Homo sapiens* CNNM1-4), plus at least one protein from every eukaryotic proteome, including the previously characterized CNNM proteins from *Saccharomyces cerevisiae* (MAM3) and *Medicago truncatula* (*Mt*CBS1) (Figure 1B). On the other hand, Clade B includes *Ot*CNNM2 and exclusively other proteins from photosynthetic species, or prokaryotic species. Unlike members of Clade A, protein sequences in Clade B have a CorC-HlyC ion transport domain. 

### 2.2. CNNM Proteins Affect Whole-Cell and Subcellular Magnesium Content

Given that Mg^2+^ cannot travel freely across biological membranes [3], the activity of the Mg^2+^ transport system directly determines how Mg^2+^ is allocated and accumulated within cells. Conflicting reports about the roles of CNNM proteins in Mg^2+^ homeostasis have been published [21]. Given the fact that *Ostreococcus tauri* has a single homolog in each of the two clades, it represents an excellent model to study the contributions of both types of CNNM proteins to Mg^2+^ homeostasis. We cloned full-length gene versions under the control of the constitutive HAPT promoter [22] and that containing a C-terminal STREPII tag. We then transfected these overexpressing vectors, validated transgene expression by Western blot, and selected lines with high protein levels for CNNM1 (C1.1, C1.4, C1.6) and CNNM2 (C2.2, C2.5, C2.6. Appendix A). We then explored the effect of overexpressing CNNM proteins on overall cellular Mg^2+^ content using a well-described in vitro luminescent assay to measure Mg^2+^ [3] in whole-cell extracts. Interestingly, our results reveal a drastic reduction of 10–15% of overall Mg^2+^ content in the overexpressing lines compared with the parent line (Figure 2A). This confirms a role for CNNM proteins in [Mg^2+^] accumulation consistent with outward transport.

However, whole-cell Mg^2+^ content is dictated by transport over the plasma membrane between the cytosol and extracellular space. To reveal differential transport from the cytosol into intracellular organelles, we differentially lysed the plasma membrane with the non-ionic detergent digitonin, releasing the cytosolic Mg^2+^ content [23]. Upon extracting Mg^2+^ from the resulting pellet, containing all intracellular-membrane-bound compartments, we found that the overexpression of CNNM proteins led to an increase of up to 20% in the accumulation of Mg^2+^ (Figure 2B). These results indicate that while increased CNNM activity lowers the overall cellular content of Mg^2+^, a larger percentage of it is in the organelles, in turn meaning that the cytosolic concentrations must be depleted. Although structural changes such as differential organelle size could affect these measurements, our results indicate that cells might compensate against low cytosolic Mg^2+^ by storing more in the organelles. Regardless, our results confirm that the abundance of representative proteins from both clade A and clade B of CNNM proteins directly affect subcellular Mg^2+^ homeostasis.

### 2.3. CNNM Proteins Affect Circadian Rhythmicity

Our previous work established that intracellular Mg^2+^ levels are important for the circadian rhythmicity of cells [3]. We then sought to test whether the observed changes in Mg^2+^ levels upon overexpression of CNNM1 or CNNM2 proteins translated directly to changes in circadian gene expression. The background line we used to generate CNNM overexpression lines expressed a fusion protein of the TTFL clock gene Circadian Clock Associated 1 (CCA1) with firefly luciferase (LUC) [22]. The lines enabled us to study the effect of CNNM overexpression on the parameters of TTFL gene expression rhythms. Our results show that under constant light conditions (LL), CCA1-LUC rhythms are significantly lengthened upon the overexpression of either CNNM1 (Figure 3A–D) or CNNM2 (Figure 3E–H). This indicates that both CNNM proteins directly impact on circadian timekeeping. 

### 2.4. The Effect of CNNM2 but Not CNNM1 on the Circadian Period Is Synergistic with Inhibition of Mg^2+^ Transport

Previous studies have established cell treatments that inhibit or alter Mg^2+^ transport, such as Cobalt(III)hexammine (Co(NH_3_)_6_^2+^, CHA) [3,15]. CHA is known to interfere and specifically block Mg^2+^ transport, leading to severe phenotypes as abolished Mg^2+^ rhythmicity [3]. To test whether the circadian defects observed in overexpressing lines of CNNM proteins (Figure 3) are directly caused by the differential accumulation of Mg^2+^ in these lines (Figure 2), we next treated the CNNMox lines with a wide range of [CHA] to test the sensitivity of the clock compared with the parent line. CHA treatments led to dose-dependent circadian period lengthening in the parent line (Figure 4A), consistent with previous results [3], as well as in the overexpression lines (Figure 4B–D). The effect of CHA on circadian rhythms in a line overexpressing CNNM1 was similar to that observed in the parent line. However, the effect of the overexpression of CNNM2 on circadian period was strikingly synergistic with CHA treatments. The synergy of CHA and CNNM2 overexpression on lengthening circadian rhythms implies that both treatments influence the clock through two separate cellular mechanisms. This result indicates that differential subcellular localization and/or the different domain architecture of Clade A and B of CNNM proteins leads to a differential sensitivity to the Mg^2+^ transport blockage.

### 2.5. CNNM Proteins Protect Cell Proliferation Rates against Magnesium Depletion

Given the observed changes to subcellular Mg^2+^ homeostasis and the circadian clock, we hypothesized that the overexpression of CNNM proteins could alter physiologically relevant properties of cellular life. We therefore tested whether the known deleterious effects of Mg^2+^ deprivation on cell proliferation were differential in CNNM overexpression versus parent lines. Cells were subjected to a four-order magnitude range of extracellular Mg^2+^([Mg^2+^]_e_), and the growth rate was measured by cell counting for 8 days. Interestingly, when compared with the parent line, the overexpression of either CNNM1 or CNNM2 can protect against extremely low extracellular Mg^2+^ (micromolar range, Figure 5A–E). However, only the overexpression of CNNM1 led to increased cell proliferation at any of the Mg^2+^ concentrations tested. The observed phenotypes in cell proliferation were not linked to clock rhythms; CCA1 rhythms were the same for all lines in all conditions (Appendix A). Although these data are consistent with the Mg^2+^ transport activity of CNNM proteins, it contradicts the hypothesis that CNNM proteins mediate Mg^2+^ extrusion out of the cells only: an efflux protein would exacerbate the effect of low extracellular Mg^2+^ on growth. This could mean that these proteins can mediate both Mg^2+^ efflux and influx, depending on the physiological and environmental status and the direction of gradients over a biological membrane.

## 3. Discussion

Our combined results show that CNNM proteins contribute to clock function, presumably through a role as a Mg^2+^ transport protein. Our fractionation data suggest that CNNMs are responsible for Mg^2+^ efflux in normal physiological conditions. This is consistent with reports on the direction of transport both for CorC proteins [24] and CNNM proteins [25]. However, CNNM overexpression rescues cell proliferation in low-[Mg^2+^]_e_ conditions compared with the parent line. This appears inconsistent with the fractionation data that shows these lines have a lower cytosolic Mg^2+^ level. It is known that a reduction in extracellular Mg^2+^,[Mg^2+^]_e_, causes a reduction in intracellular Mg^2+^,[Mg^2+^]_i_ [3]. Therefore, we would expect the overexpression of CNNM to compound the deleterious effect of Mg^2+^ depletion. However, because of the radical decrease in [Mg^2+^]_e_ from the normal physiological conditions for these cells (50 mM), the [Mg^2+^]_e_/[Mg^2+^]_i_ ratio will be different, and the gradient will be reversed from [Mg^2+^]_i_ < [Mg^2+^]_e_ to [Mg^2+^]_i_ > [Mg^2+^]_e_. We hypothesize that CNNM proteins can flip the direction of transport depending on this gradient, as reported previously (directional flipping [26]), which could explain the paradoxical results of the growth tests compared with fractionation assays. 

Our phylogenetic analysis shows two distinct clades for CNNM proteins, with one *Ostreococcus* protein present in each clade, suggesting a distinct evolutionary history for these two proteins. Clade A (in which the *Ostreococcus* CNNM1 protein is present) contains proteins known to be located at the plasma membrane (such as the human proteins and MGR4-7 *Arabidopsis* proteins), and the vacuolar membrane (for the MAM3 yeast protein and MGR1-3 *Arabidopsis* proteins [27,28,29]). In contrast, Clade B (with the *Ostreococcus* CNNM2 protein), only includes proteins from prokaryotic or photosynthetic eukaryotic species. Additionally, the two *Arabidopsis* proteins in Clade B are known to be located on chloroplast membranes (MGR8-9) [30]. Although CNNM2 does not have a chloroplast targeted transit-peptide (as analyzed with TargetP-2.0 [31,32]), we hypothesize that CNNM2 (and other Clade B members) is localized at the chloroplast, as expected from the phylogenetic tree. The evidence presented in the phylogenetic tree is corroborated by searches using the online webtool AlgaeFUN [33]. Searches with AlgaeFUN of the two *Ostreococcus* proteins indicate that they belong to two separate orthogroups, which correspond to the two Clades identified in Figure 1B. From this evidence, we can hypothesize that the eukaryotic proteins in Clade A and Clade B were acquired separately from two different prokaryotic ancestors. For example, it is possible that, for clade B proteins, the ancestral eukaryotic protein was acquired through the endosymbiosis, which gave rise to the chloroplast, and the genes encoding these proteins were subsequently transferred to nuclear genomes. Spatially separated from CNNM1, efflux would therefore occur across the chloroplast membrane. The change in total [Mg^2+^]_i_ observed is consistent with this hypothesis, as a change in chloroplast [Mg^2+^] would have a knock-on effect on total [Mg^2+^]_i_. Indeed, the effect of an organellar Mg^2+^ transporter on total [Mg^2+^]_i_ has previously been reported using overexpression of the mitochondrial human MRS2 protein, which increased total cellular Mg^2+^ [34]. The identification of CNNM2 as a chloroplast efflux transport protein in the green lineage is an important step forward in understanding the mechanisms underlying rhythmic Mg^2+^ fluxes, and complements the recent identification of *Os*MGT3 by Li et al. (2020) as the mediator of rhythmic influx across chloroplast membranes in rice.

A recent study revealed separate functions for human CNNM proteins in Mg^2+^ efflux and influx into cells. However, the influx activity depended on interaction with the TRPM7 protein, which is a Mg^2+^ channel and protein kinase only found in mammals, whereas the efflux activity does not [21]. The absence of TRPM7 outside mammals means that this second function cannot be conserved across species, nor contribute to our results. Another complexity in mammals is that CNNM proteins can interact with phosphatase of the regenerating liver (PRL) proteins [19]. Both PRL and CNNM genes are rhythmically expressed in a variety of mice tissues, and PRL2-KO mice exhibited an altered circadian clock, through a role for PRL2 in circadian energy and metabolism by regulating [Mg^2+^]_i_. However, again, *Ostreococcus* does not contain any homologs of PRL proteins.

Our study reveals CNNM protein functions in the absence of the complexity that exists in mammals. That said, we show that even without these additional proteins, the function of CNNM proteins is complex and differential between Clade A and Clade B proteins. Based on the evidence presented here and in previous studies, it is clear that (rhythmic) Mg^2+^ transport needs to be studied at the subcellular level by combining the study of influx and efflux transport proteins at the separate biological membranes.

## 4. Materials and Methods

### 4.1. Phylogenetic Analyses

Homologs of characterized CNNM proteins were searched for in the predicted proteomes of model species representative of various eukaryotic taxonomic groups, including *Ostreococcus*. To identify homologs of characterized CNNM transport proteins, we manually used the BLAST webpage interface (NCBI) [35]. BLAST searches with the human CNNM2 protein sequence (Uniprot protein Q9H8M5), using BlastP with default search parameters against the reference proteins database (refseq_protein) by specifying the organism queried. Searches used the blastP (protein–protein BLAST) algorithm. We did not download any proteins with E-values lower than the threshold, and removed all redundant isoforms and partial sequences. Proteins were renamed for phylogenetic analysis to include the organism name and a protein number or protein name if previously published (Appendix A). Additionally, the CorA and CorC proteins from *Methanoculleus thermophilus* and *Gloeomargarita lithophora*, and the CBS1 protein from *Medicago truncatula*, were added to this list. All curated protein sequences were aligned using MUSCLE in MEGA 11.0.11 using default settings [36]. Protein alignments were used to build maximum likelihood (ML) trees (100 bootstraps), using the Jones–Taylor–Thorton (JTT) substitution model and the nearest neighbor interchange (NJJ) substitution model. 

### 4.2. Ostreococcus tauri Lines

*Ostreococcus* cultures were grown for 6–7 days in artificial sea water (ASW) under 12/12 h light/dark cycles at 20 °C, under enriched blue light (183 Moonlight Blue Filter), as previously described [37]. New transgenic lines overexpressing *OtCNNM1* (*ostta05g01490*) and *OtCNNM2* (*ostta11g01030*) genes were PCR-amplified using specific primers incorporating the STREP tag sequence (Appendix A). Then, fragments were digested with AvrII (NEB) and ligated into the pOTOX vector [22]. Genomic transformation was conducted as previously published [38] and performed under the CCA1-LUC background line [22]. Positive colonies were selected with ClonNAT antibiotic (Nourseothricin, Jena Bioscience) and protein samples were validated by immunoblotting using an anti-STREP antibody (Strep-II antibody, abcam ab184224). 

### 4.3. Plate Assays

Seven-day-old *Ostreococcus* cultures were diluted at a 1:3 dilution rate and mixed with 0.2 mM D-luciferin. Then, 90 µL of this solution was added to wells in a 384-well plate (Greiner) and imaged in a TriStar2 luminescent plate reader (Berthold) under 2 μmols m^−2^ s^−1^ of blue light (183 Moonlight Blue Filter; Leefilters) under LL. To test the effect of CHA (Sigma), wells were filled with 81 µL of diluted culture and mixed with 9 µL of each 10x [CHA] (or water). For washout treatments for low-Mg concentrations, 80–90 µL media from each well was carefully replaced with 80–90 µL of treated media with luciferin (or ASW + luciferin), avoiding disturbing the cells. Results were analyzed and plotted using GraphPad Prism v9 or with BioRender. Circadian parameters were quantified with BioDare2 [39].

### 4.4. Cell Fractionation and Mg Quantification

Cell pellets were harvested from 1-week-old cultures and resuspended in Buffer 1 (50 mM Hepes-HCl pH 8, 150 mM NaCl). For whole-cell samples, cell suspensions were mixed while vortexing with equal volumes of 2x Buffer 1 + 1% Triton X-100; then, the supernatant was collected after centrifugation at maximum speed. For organelle fractions, suspensions were mixed with 2x Buffer 1 + 0.01% Digitonin (Sigma). After 5 min incubation at 4 °C and centrifugation at 4000 rpm for 15 min, the organelle fraction was obtained after discarding the resulting supernatant and resuspending pellets in Buffer 1 + 0.5% Triton X-100. 

The quantification of intracellular Mg^2+^ was performed using a luminescent plate assay, as previously described [3], with a slight alteration: an assay buffer with 50 mM Hepes-HCl pH 8, 1 mM luciferin, 0.05 mg·mL QuantiLum (Promega), and 1 mM ATP was used [3]. Data from at least 3 technical replicates for each time point were plotted.

### 4.5. Growth Assays

*Ostreococcus* growth rates were quantified by counting changes in the cell number using a hemocytometer with a light microscope at 40× magnification. The same starting cell density from 1-week-old cultures was resuspended and incubated in different low-Mg media under LL conditions, and the cell number was counted every 2 days. Data from at least 5 technical replicates for each time point were plotted.

## Figures and Tables

**Figure 1 ijms-24-02273-f001:**
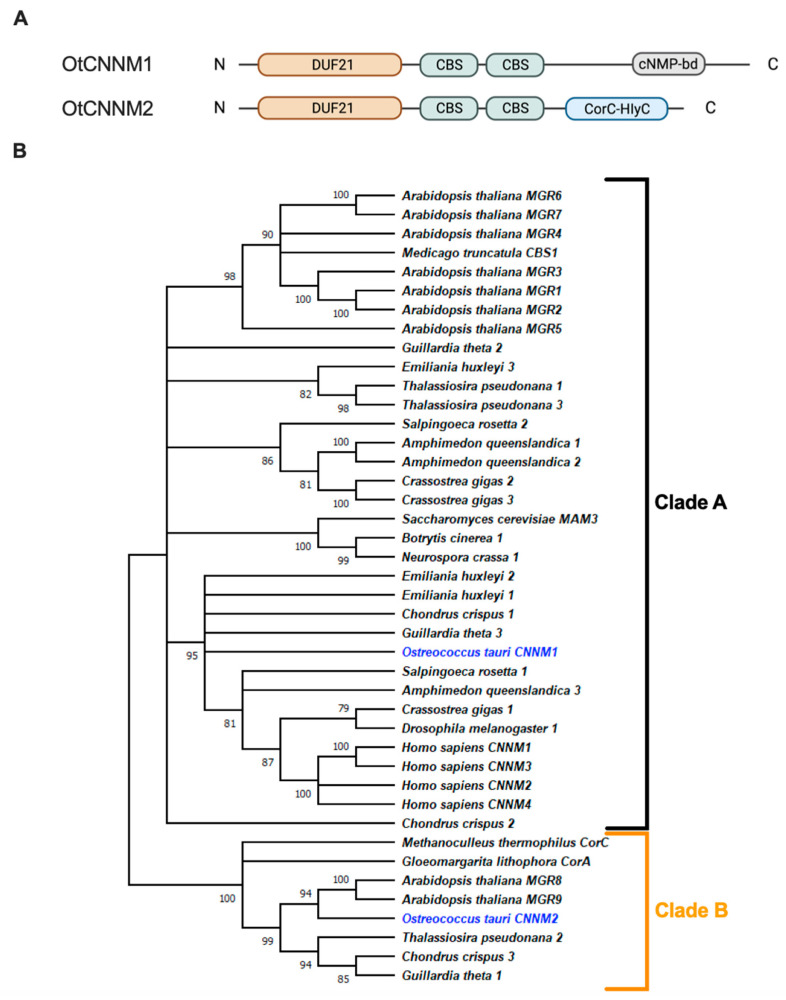
Structure and ancestry of two *Ostreococcus* CNNM proteins (marked blue). (**A**) Schematic diagram of the structure of the CNNM proteins found in *Ostreococcus tauri*. (**B**) Maximum likelihood (ML) phylogenetic tree (100 bootstraps) of whole protein sequences for CNNM homologs.

**Figure 2 ijms-24-02273-f002:**
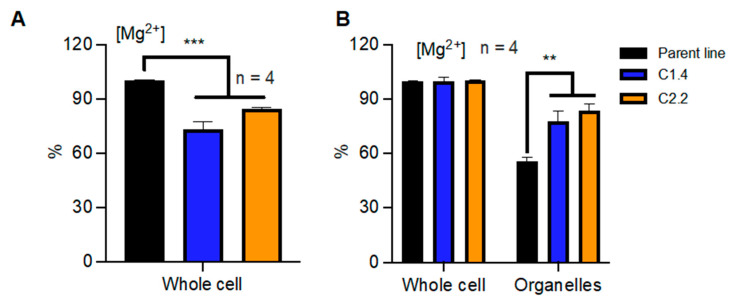
CNNM proteins affect subcellular magnesium homeostasis. Relative quantification of total [Mg^2+^] in the CNNM-overexpressing lines for whole-cell extracts (**A**) and organelle fractions (**B**). Data are relative to [Mg^2+^] in the parent line in panel (**A**), and relative to [Mg^2+^] in whole cell extracts in panel (**B**). Data are presented as the mean ± SEM. n = technical replicate number. ** = *p* < 0.01, *** = *p* < 0.001; Student’s *t*-test.

**Figure 3 ijms-24-02273-f003:**
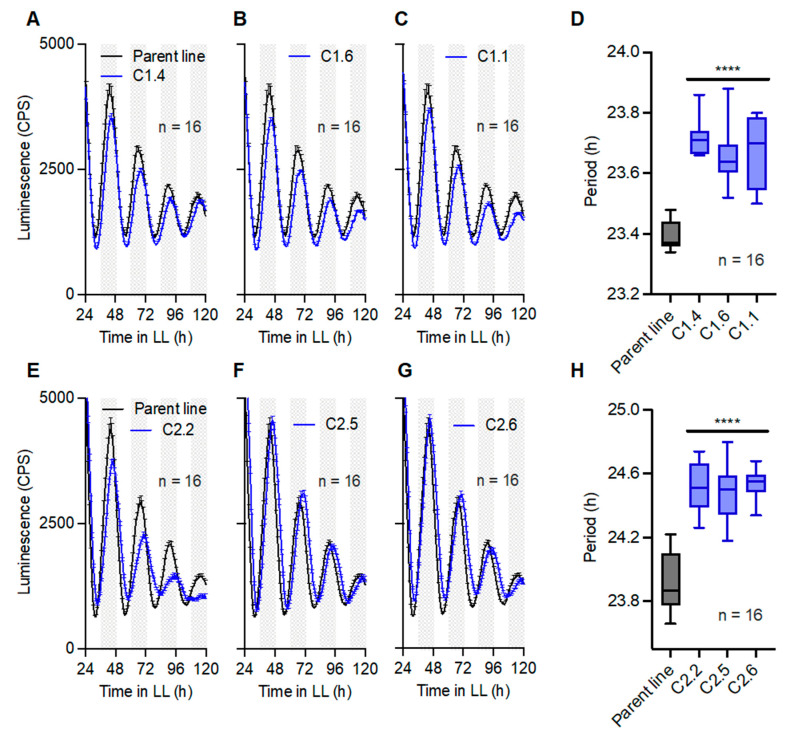
CNNM proteins affect circadian clock gene expression rhythms. Luminescent CCA1-LUC rhythms and period analyses under LL for lines overexpressing CNNM1 (**A**–**D**) and CNNM2 (**E**–**H**). Luminescent data are represented as mean ± SEM. Period data are shown as boxplots, where middle lines indicate the median, lower and upper boundaries represent the 25th and 75th percentiles, respectively, and error bars show data distribution from lowest to highest value. n = technical replicate number. Grey areas represent subjective night periods. **** = *p* < 0.0001; statistical analyses are compared with the parent line; Student’s *t*-test. CPS = Counts per second, h = hours.

**Figure 4 ijms-24-02273-f004:**
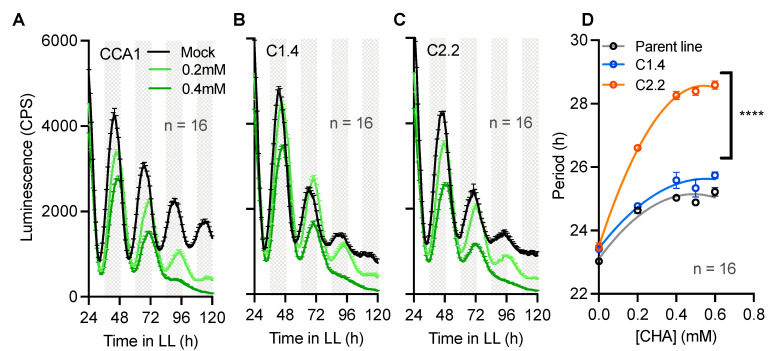
CNNM2 activity acts synergistically with magnesium transport inhibition. CCA1-LUC luminescent traces in response to CHA treatments under LL for CCA1 (parent line, (**A**)), CNNM1ox (**B**), and CNNM2ox (**C**) and period analyses to a range of [CHA] (**D**). Data are presented as the mean ± SEM. n = technical replicate number, h = hours. **** = *p* < 0.0001; statistical analyses are compared with the parent line; two-way ANOVA.

**Figure 5 ijms-24-02273-f005:**
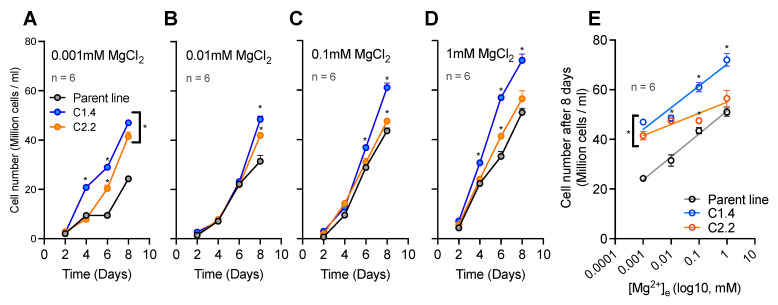
CNNM activity rescues cell proliferation under magnesium depletion. Cell growth time series under LL for the parent line, CNNM1ox, and CNNM2ox subjected to low-Mg media: 0.001 mM (**A**), 0.01 mM (**B**), 0.1 mM (**C**), and 1 mM (**D**) MgCl_2_. (**E**) Cell counting after 8 days of exposure to low-Mg media. Data are presented as the mean ± SEM. n = technical replicate number. * = *p* < 0.0001; statistical analyses are compared with the parent line; two-way ANOVA.

## Data Availability

Not applicable.

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
