# Peer review of "Homologs of Ancestral CNNM Proteins Affect Magnesium Homeostasis and Circadian Rhythmicity in a Model Eukaryotic Cell"

_ijms, 2023, doi:10.3390/ijms24032273_

Round 1
Reviewer 1 Report
This work presents a relevant characterization of the two CNNM proteins in Ostreococcus tauri focusing on their control over Mg2+ homeostasis, circadian clock regulation and cell survival. The paper is well written, easy to read and follow. Nevertheless, the following minor revisions may improve both the content and presentation of the results:
(1) I suggest avoiding abbreviations in the title and abstract of the paper and replace CNNM by cyclin M protein.
(2) In Fig1 legend, remove an extra dot and remove Created with Biorender. If necessary include this information in Materials and methods with the corresponding citation.
(3) Fig2. C1.4 and C2.2 are never introduced in the main text only in Fig S1. I suggest introducing the names of the different overexpressing lines for OtCNNM1 and OtCNNM2 in the main text. For instance, C1.4 OtCNNM1ox1, C1.6 OtCNNM1ox2 and C1.1 OtCNNM1ox3; C2.2 OtCNNM2ox1, C2.5 OtCNNM2ox2 and C2.6 OtCNNM2ox3. C2.6 is also misplaced in Fig S1. Only one line overexpressing OtCNNM1 and OtCNNM2 is shown whereas for rhythmicity in all lines is analysed. Is the effect of Mg2+ concentration similar in all lines?
(4) Fig2 legend, replace p < 0,01 with p < 0.01.
(5) Fig2 B left shows “Whole cell” identical to Fig2 A whereas significant changes are shown in Fig2A and no changes in Fig2 B. It is not clear what Fig2 A and Fig2 B left represent. Please clarify.
(6) Fig3. There is an extra space in affect (affec t). Replace p < 0,01 with p < 0.01.
(7) line 158 replace “overexpression lines” with “overexpressing lines”
(8) Fig4 The color selection is confusing. In A, B and C black, blue and orange represent Mock, 0.2mM and 0.4mM wheres in D the same colors represent Parent line, C1.4 and C2.2. I suggest to change the colors in D. In this case again a single line is analysed. Are there any similar data for the other lines?
(9) FigS2. Please indicate what the vertical dotted red line represents (transition to constant light?). How are these values related to those presented in Fig3 A and E? In Fig 3A a clear amplitude decrease in C1.4 is observed in the second subjective night whereas in FigS2 A an B and apparent amplitude increase is observed in C1.4 (over 6000 CPS) with respect to CCA1 (below 6000 CPS). Please clarify.
(10) Fig5. I personally would understand this figure better if it was presented D, C, B and A, in increasing MgCl2 concentration similar to E. Similar to the response to CHA concentrations, no synergistic effect is observed in the overexpression of OtCNNM1. The blue and black lines in Fig5 E are parallel (similar to Fig 4 D blue and black lines). In contrast to OtCNNM2 overexpressing line that presents an amplifying divergence in cell survival as Mg2+ decreases. sThis suggests that OtCNNM2, specific to photosynthetic organisms, induces synergistic responses to Mg2+ whereas OtCNNM1 induces a different but “parallel” global cellular state with respect to the parent line. Different but equivalent (parallel) responses are observed in both C2.2 and the parent line. The C2.2 line present also the highest increase in organelle Mg2+ (chloroplast) content with respect to the parent line. A discussion in this respect would be appreciated by this reviewer.
(11) Line 197 Indicate “extracellular Mg2+ concentration, [Mg2+]e”. This notation is introduced later on line 199. This notation is also used in Fig5 whithout being previously introduced.
(12) Ostreococcus tauri is considered the eldest sister in the green lineage being specially relevant in evolutionary studies. In this respect, I would have appreciated a more detailed discussion beyond the phylogenetic tree presented in Figure 1 B. For instance, using the Funtree application in AlgaeFUN (https://greennetwork.us.es/AlgaeFUN/) it can be inferred that OtCNNM1 and OtCNNM2 belong to two different orthogroups (group of ortholog genes) suggesting a different evolutionary history in the green lineage. OtCNNM1 belongs to an orthogroup where the Arabidopsis genes AT4G33700, AT2G14520, AT1G47330, AT5G52790, AT4G14230, AT4G14240 and AT1G03270 are located. All the proteins codified by these gene are located in the plasma membrane according to the functional annotation in TAIR. On the contrary, OtCNNM2 belongs to a distinct orthogroup where AT3G13070 (MRG8), AT1G55930 (MRG9) and AT4G00925 are located. MRG8 and MRG9 are characterized in Zhang 2022 and are located in the chloroplast envelope and AT4G00925 protein is located in the mitochondrion. This supports the results presented in the paper related to the synergistic and therefore more relevant role of OtCNNM2 whereas OtCNNM1 overexpression solely induces a different “parallel” state of the cell.
Reviewer 2 Report
This interesting study concerns proteins that regulate Mg2+ flux into and out of cells, and their effects upon circadian rhythms. Previous studies have identified circadian rhythms in the concentration of Mg2+, but the mechanisms that lead to these rhythms, and their downstream effects need more exploration. For this reason, this paper represents an informative contribution to the field. The study used Ostreococcus tauri, a model alga that has become established for certain types of studies of circadian rhythms, as an experimental model.
Overall, the study is sensible and well-interpreted. I have a few questions and concerns:
1. The detail concerning statistical analysis and its representation on the figures needs improvement. When such analysis is used, it does need to have proper explanation. In Fig. 2 legend, no explanation is provided for the tests that were used, and it is not explained what the error bars represent. In Fig. 3 legend, no explanation is provided for the tests that were used, and the error bars are not explained (e.g. the error bar type might be different from Fig. 2, because here boxplots are used). Please can the authors be specific about whether n = 16 means 16 wells on a microplate, or whether it means 16 independent experimental trials. In Fig. 4 legend, no explanation is provided for the tests that were used, and it is difficult to tell from Fig. 4D what is being compared to what with the ****. The authors might need to think of a different way to show this comparison. What algorithm was used to fit the curves to the data (is the curve type appropriate, given that the curve implies that in the parent line, the period was reduced at the greatest concentration of CHA?). In Fig. 5 legend, no explanation is provided for the tests or error bars.
2. In Figure 2, I was puzzled by why the Mg2+ levels are different in Fig. 2A, but this doesn’t seem to also be the case for the left-hand part of Fig. 2B, which is also whole-cell preparations. I suspect they are slightly different experiment types (if they are not, the authors’ results are inconsistent). Please can the authors check whether or not they are different experiment types and, if this is the case, add some clearer explanation of the nature of the difference. If it confused this reviewer, it would probably confuse other readers also.
3. In a number of places, the manuscript claims to study “ancestral” forms of CNNM proteins. This does not seem to be an appropriate form of words, because Ostreococcus is a contemporary species and not something ancestral, e.g. from the fossil record. Whilst it diverged from multicellular plants a long time ago, it will have followed its own evolutionary trajectory, so the proteins in Ostreococcus might not be the same as a last common ancestor. I think this claim needs to be revised throughout the manuscript, because it's currently over-interpretation.
4. Line 193-194 needs some nuance. The results do not prove that CNNM proteins act upon the clock through Mg2+. They support the idea that could be the case- but there could be something unknown occurring here also, e.g. CNNM proteins conducting other ions.
Minor
Should the species names in Fig. 1 be in italic (Latin names)?
Abstract: Some parts of the abstract need to be toned down. Are circadian rhythms ubiquitous in all organisms (have “all organisms” been studied for circadian regulation?). Are Mg2+ transport proteins “hotly debated” (I did not have this impression from the Introduction).
It might be worth quickly checking whether CNNM2 in Ostreococcus has a predicted chloroplast transit peptide, given the paragraph at line 207. It might not if it is in the outer envelope, but if it was deeper inside the chloroplast, it is possible that it would.
Line 34: “studies” should be “study”.
Line 35: “in” should be “with”.
Line 44: Are transport proteins “alongside” the cellular membranes, or embedded within them?
Line 50: Is there a reference to support this statement, or is it a hypothesis from the authors? It would be good to have clarification.
Line 57: procaryotic -> prokaryotic.
Lines 90-94: Please consider whether this is interpretation / speculation that is better suited to the Discussion than the Results.
Line 204-206: This is an interesting hypothesis (directional flipping). Are there any other examples of this from the literature, for ion transport proteins, that could be used to support the hypothesis?
